# The *Drosophila* Connectome as a Computational Reservoir for Time-Series Prediction

**DOI:** 10.3390/biomimetics10050341

**Published:** 2025-05-21

**Authors:** Leone Costi, Alexander Hadjiivanov, Dominik Dold, Zachary F. Hale, Dario Izzo

**Affiliations:** 1Advanced Concepts Team, European Space Research and Technology Centre, European Space Agency, 2201 AZ Noordwijk, The Netherlands; 2Netherlands eScience Center, 1098 XH Amsterdam, The Netherlands; 3Faculty of Mathematics, University of Vienna, 1090 Vienna, Austria

**Keywords:** reservoir computing, biomimetics, *Drosophila*, neural networks, bio-inspired computing

## Abstract

In this work, we explore the possibility of using the topology and weight distribution of the connectome of a *Drosophila*, or fruit fly, as a reservoir for multivariate chaotic time-series prediction. Based on the information taken from the recently released full connectome, we create the connectivity matrix of an Echo State Network. Then, we use only the most connected neurons and implement two possible selection criteria, either preserving or breaking the relative proportion of different neuron classes which are also included in the documented connectome, to obtain a computationally convenient reservoir. We then investigate the performance of such architectures and compare them to state-of-the-art reservoirs. The results show that the connectome-based architecture is significantly more resilient to overfitting compared to the standard implementation, particularly in cases already prone to overfitting. To further isolate the role of topology and synaptic weights, hybrid reservoirs with the connectome topology but random synaptic weights and the connectome weights but random topologies are included in the study, demonstrating that both factors play a role in the increased overfitting resilience. Finally, we perform an experiment where the entire connectome is used as a reservoir. Despite the much higher number of trained parameters, the reservoir remains resilient to overfitting and has a lower normalized error, under 2%, at lower regularisation, compared to all other reservoirs trained with higher regularisation.

## 1. Introduction

In the ever-expanding field of machine learning (ML), increasing performance generally involves increasing model complexity, which in turn lengthens learning time [1,2,3]. This is typically achieved in one of two ways: either by adding more tunable parameters in neural networks (NNs), such as Transformers, CNNs, or LSTMs [4,5,6,7], or by increasing the number of simulations in reinforcement learning and genetic algorithms [8,9,10].

Other NN architectures, such as those used in echo state networks (ESNs) [11] and liquid state machines (LSMs) [12], form a category of models referred to as reservoir computing, where most of the computation is performed by a randomly connected network of untrained synapses. Reservoir computing is a neural network paradigm that requires minimal training compared to most advanced ML models [13,14]. In a similar fashion to radial basis functions [15], reservoir computing relies on a nonlinear expansion of the dimensionality of the input data, which can then allow the problem to be solved linearly by a simple fully connected layer known as the readout. A notable difference between the two approaches is the recurrent nature of reservoirs, which adds a temporal buffering behavior to the system [16,17]. This characteristic represents a fundamental advantage in the task of time-series prediction, in which reservoir computing performs exceptionally well [18,19], even in the case of chaotic time-series [20,21].

While the original concept of a reservoir consisted of virtually simulated pools of neurons that act as nonlinear expansions and temporal buffers for time-series, physical implementations spanning several scientific fields have already been implemented, with examples including physical reservoir computing [22,23], photonic reservoir computing [19,24], and even memristor reservoir computing [25,26]. Broadly speaking, depending on the neuronal model used as the activation function [27,28], reservoir computing can be subdivided into liquid state machines, using spiking neurons, echo state networks, based on classical firing-rate neuron models, and time-delay reservoirs, introducing a temporal delay to the neuron dynamics.

The standard practice when creating a reservoir involves randomly initializing a desired number of neurons and synapses, adhering to a specified sparsity, spectral radius, and distribution of non-zero weights [29,30]. While reservoir size, spectral radius, and sparsity significantly influence performance, other parameters often receive limited attention [21,31].

Despite these standardized practices, we still lack an understanding of what constitutes a good reservoir. One promising line of research focuses on exploring physical systems with reservoir-like properties by using various combinations of materials, structures, and processes [32]. For instance, $SnO_{x}$-based physical reservoirs have been shown to exhibit key properties observed in biological systems, such as long-term potentiation (LTP) and depression (LTD) [33]. In addition, polymer electrolyte-gated $MoS_{2}$ transistors [34] and stochastic memristive devices in the form of a 3D filament matrix [35] have demonstrated reservoir-like behavior suitable for tasks relying on memory and the ability to learn temporal dynamics, where they maintain low error over relatively long prediction horizons for tasks such as chaotic time-series prediction and high accuracy on tasks such as spoken digit recognition.

Nature, on the other hand, has evolved neuronal systems over millions of years in the form of animal brains with a vast diversity of capacities, functions, and specializations [36,37]. The example given by nature is often taken as an inspiration in the field, which in recent years has seen several attempts at bio-inspired and biologically plausible reservoir computing [38], from the imitation of neuronal plasticity [39] to the artificial implementation of the role of both neurons [40] and smaller biomolecules [41].

One study has taken a more drastic approach involving moving away from bio-inspiration and towards biomimicry by mapping the connectivity of connectomes of different species to a reservoir, in an attempt to investigate the performance of the former [42]. However, that study was limited both by the limited range of investigated hyperparameters and by the size of the investigated connectome, as at the time there was no reported case of a complete scan. In particular, whereas the reported result of connectome-based topology and random weights seems to imply a performance at best comparable to the standard fully randomized reservoir, the attempt at also simulating the synaptic weights was unsuccessful, further lowering performance for the heterogeneous connectivity case and completely failing for the homogeneous one. Following that study, connectome topologies have been proposed as an additional constraint on top of standard practice [43].

Similarly, both the topology and the distribution of synaptic weights of several partial connectomes, including humans, have been shown to play a fundamental role in affecting the memory capacity of connectome-based reservoirs. On one hand, the presence of small-world organization in biological connectomes is shown to increase the resulting memory capacity due to extreme robustness and effective communication [44]. This is achieved with a few hyperconnected neurons that create redundant clusters paired with long-range neurons that quickly transfer information between different clusters. Similarly, Suarez et al. [45] have shown how connectome-based reservoirs perform better near the edge of chaos, thanks both to their topological structure and the distribution of synaptic weights. Whereas these works successfully show that a biomimetic approach to generating reservoirs can increase their memory capacity, they have been performed based on partial connectomes of limited sizes, and they do not directly show how specific reservoir features affect the predicted trajectories when used as time-series predictors.

Nevertheless, thanks to the *FlyWire* project, the complete connectome of a fruit fly brain has been recently published [46]. This represents an important milestone in neuroscience since it is the first complete scan of a relatively large brain. The only other documented complete scan of a connectome is of the *Caenorhabditis elegans*, with only 302 neurons [47], which has been investigated as an AI model [48] but contains too few neurons to be used for a thorough investigation within the field of reservoir computing. This recent scientific advancement provides a unique opportunity to analyze the topology and structure, as well as the role of synaptic weights, of an animal’s neural connectivity up to the scale of the entire connectome.

In this work, we explore the possibility of using the topology and synaptic connectivity of the scanned *Drosophila* connectome as a computational reservoir for time-series prediction, comparing its performance to state-of-the-art implementations. Whereas the topology of limited parts of the *Drosophila* connectome, namely the olfactory system [49], has already yielded good preliminary results, the implementation of different sparsities between the biomimetic architecture and the control case, in addition to the limited size of the study itself, prevented a deeper understanding of the relationship between the connectome’s features and the observed increase in prediction performance. To the authors’ knowledge, this is the first reported study that investigates and characterizes the combination of topology and synaptic weights for an entire connectome and showcases its performance as a time-series predictor. Compared to previous works on connectome-based reservoirs for time-series prediction, we propose a much wider study, both regarding the reservoir’s size, including several different subsets of increasing size up to the entire connectome, and the explored hyperparameter space. We also investigate hybrid architectures, which are only partially based on the biological data, in order to characterize the individual effects of connectome topology and synaptic weights, respectively. To achieve this, we perform time-series predictions with an increasing forecast horizon, utilizing multiple different trajectories of a notoriously chaotic nonlinear time-series: the three-body problem [50]. Not only does this problem represent a historical milestone in the investigation of Lagrangian determinism, but it is also characterized by a greater challenge than other chaotic time-series often used for benchmarking since the network is required to predict three time-series simultaneously (the x,y, and *z* positions of the body). Moreover, it follows recent interest in generative techniques for the *n*-body problem [51]. Our results demonstrate that the performance of the fly connectome is noticeably different from that of classical ESNs. Notably, reservoirs based on the fly connectome exhibit reduced overfitting in scenarios where overfitting plays a significant role [50].

## 2. Materials and Methods

### 2.1. Connectome Extraction

In order to represent the fly connectome as a reservoir, the data of both neurons and synapses were extracted using the CAVE client Python interface provided by the library *caveclient 6.4.1*. Neurons and neuronal classes were extracted simultaneously, saving the unique ID, the x,y, and *z* spatial positions, and the cell class of every single neuron. For reference, Table 1 outlines all the different classes and their labels and briefly describes their function. With the extracted information, it was possible to visualize the *Drosophila* connectome and to group its neurons according to their properties, as shown in Figure 1.

In order to create a reservoir, we retrieved all the synapses that had a given neuron as a pre-synaptic neuron for every neuron previously collected. Queries of the database returned the following information:The unique IDs of the pre-synaptic and post-synaptic neurons;The Cleft score: a measure of how well-defined the synaptic cleft is;The Connection score: a measure of the size of the synapse;The probability of the synaptic neurotransmitter being Gamma-Aminobutyric Acid, Acetylcholine, Glutamate, Octopamine, Serotonin, or Dopamine.

The first pre-processing of the data was performed by applying a synapse quality filter: a simple threshold-based filtering of the synapses. First, synapses in which the post-synaptic neuron was not within the original set of classified neurons were deleted. Then, all synapses with either a Connection score or a Cleft score below a given threshold were also deleted to avoid false positives. Following the guidelines of the original work [46], we selected these thresholds to be 50 for the Cleft score and 100 for the Connection score. Whereas there are few synapses with a Connection score as high as $10^{4}$, most of them are between $10^{2}$ and $10^{3}$.

In order to transform the biological data into a connectivity matrix suitable for a reservoir, we had to determine the weights of the connections between neurons. Unfortunately, the provided database did not provide physiological information, but only anatomical information. Since synaptic strength can only be inferred from direct measurements, in this work, we used the Connection score as an estimate of the strength of each synapse, assuming that larger synapses lead to stronger interactions, as shown for limited neuronal pools in the literature [52,53].

Furthermore, information about the synaptic neurotransmitter was used in order to determine the signs of the weights in the connectivity matrix. Among all neurotransmitters, Gamma-aminobutyric acid is considered an inhibitory neurotransmitter, corresponding to a negative weight, and Acetylcholine and Glutamate are considered excitatory, corresponding to positive ones. Conversely, Octopamine, Serotonin, and Dopamine do not directly act as excitatory or inhibitory, but they instead modulate the resting potential of the post-synaptic neuron. Since ESNs feature a firing-rate neuronal model rather than a spiking one, they do not explicitly represent the resting potential, and thus the synapses that utilize modulatory neurotransmitters were not taken into consideration. For each synapse, we determined the neurotransmitter by taking the one with the highest probability in the database.

The result of this extraction process is a graph with 104,909 nodes and the associated connectivity matrix. Computationally, a reservoir of such a size is too expensive to run: for comparison, most studies focus on ESNs with $10^{2}$ to $10^{3}$ neurons. To obtain a smaller reservoir, we opted to retain a given number, *N*, of connected neurons, where *N* is arbitrary. To do this, we implemented two possible selection criteria: ‘most connected’ and ‘proportional’. In the ‘most connected’ case, the *N* most connected neurons were selected. However, it is important for the fairness of comparison with the control classical reservoir that the selected subset of neurons consists of a single connected graph in order to form a single reservoir of size *N* rather than two or more independent smaller ones. Therefore, only the largest connected component was considered, resulting in a reservoir of size M≤N. Then, the N−M nearest neighbors were added to obtain the desired size, prioritizing the most connected neurons among all nearest neighbors.

In contrast, the ‘proportional’ criterion attempts to preserve the overall class distribution of the connectome, selecting the *N* most connected neurons while maintaining the same proportion of each class of neurons with respect to the entire connectome. Then, to obtain a single connected reservoir, the minimum number of nodes needed to connect all selected components were added. The resulting reservoir was of size M≥N. Finally, the least connected M−N nodes were pruned from the graph, obtaining the desired reservoir size. Note that due to the requirement for a single connected subgraph, additional extraction steps were applied after the application of the selection criteria. These steps affected the final selected reservoir, which might differ significantly from the original *N* initially selected neurons.

Once a reservoir has been defined, its connectivity matrix can be extracted and used for the practical implementation of the ESN. This matrix would have a specific spectral radius, which corresponds to the largest eigenvalue of the connectivity matrix. The matrix is then rescaled with respect to its original spectral radius and multiplied by 0.99. This procedure ensures that the Echo State Property (ESP) is respected and that the reservoir operates near the edge of chaos, as indicated in previous work on time-series [49], which has been shown to improve the performance of both ESNs and biomimetic ESNs [44,45].

To characterize the topology of each extracted reservoir, three standard metrics were taken into consideration: neuron degree, clustering coefficient, and shortest path length. The degree $k_{i}$ of a neuron *i* is the number of synapses incident to it, and the degree distribution captures the frequency of each $k_{i}$ across all neurons. The local clustering coefficient $C_{i}$ of a node *i* is defined as follows:(1)$$C_{i}=\frac{2e_{i}}{k_{i}\left(k_{i}-1\right)}$$
where $e_{i}$ is the number of edges between the $k_{i}$ neighbors of neruon *i*. The shortest path length $\ell_{ij}$ is the minimum number of synapses traversed to reach neuron *j* from neuron *i*; we computed the distribution of all $\ell_{ij}$ with i≠j across the entire reservoir. To account for directionality, the clustering and degree distributions were computed separately for incoming and outgoing synapses in order to differentiate between pre-synaptic and post-synaptic neurons, and the shortest path was computed considering directionality.

### 2.2. Echo-State Network

The aim of this work is to investigate the *Drosophila* connectome as a reservoir within an ESN framework. Let us consider the equations governing an ESN (Figure 2):(2)$$x_{n}=\left(1-\alpha \right)x_{n-1}+\alpha \tanh\left(Wx_{n-1}+W^{\mathrm{in}}\left[1;u_{n}\right]\right),\;n=1,\ldots ,T$$
where *T* is the number of timesteps in the time-series, $x_{n}$ represents the reservoir state at timestep *n*, W is the reservoir weight matrix, $W^{\mathrm{in}}$ is the input weight matrix, $u_{n}$ is the input signal, and α is a scalar between 0 and 1, so that 1−α is the leakage rate.

The output readout layer is defined as follows:(3)$${y^{\mathrm{out}}}_{n}=W^{\mathrm{out}}\left[1;x_{n};u_{n}\right],$$
where $W^{\mathrm{out}}$ is the output weight matrix. The objective is to learn $W^{\mathrm{out}}$ such that ${y^{\mathrm{out}}}_{n}$ closely approximates a given target behavior (signal or label) $y_{n}$. By organizing all target readouts into a matrix $Y\in R^{N_{y}\times T}$ and all reservoir dynamics into a matrix $X\in R^{(1+N_{x}+N_{u})\times T}$, we obtain(4)$$Y=W^{\mathrm{out}}X,$$
which constitutes an overdetermined system in the unknown $W^{\mathrm{out}}$. The least squares solution, which minimizes the mean squared error (MSE), is given by(5)$$W^{\mathrm{out}}=YX^{T}{\left(XX^{T}\right)}^{-1}$$

A regularized solution, obtained via ridge regression [54], which minimizes the MSE along with a penalty on the weight magnitude, is given by(6)$$W^{\mathrm{out}}=YX^{T}{\left(XX^{T}+\beta I\right)}^{-1}$$
where β is the regularization parameter and I is the identity matrix.

### 2.3. Time-Series Generation

ESNs are often used for the classification and regression of time-series. In particular, they have been shown to have good performance even when trained on chaotic time-series [21]. In this study, ESNs are tested on the prediction of the circularly restricted three-body problem (CR3BP). CR3BP trajectories were obtained through propagating a set of initial conditions [r;v], where r and v are the initial x,y, and *z* positions and velocities, respectively. These trajectories were then generated through propagation using the *heyoka 6.1.2* library [55], considering the mass ratio parameter of the earth–moon system and a timestep of 0.1 in the normalized units of time. Each resulting trajectory consisted of 10,000 datapoints and was subdivided into three subsets: washout, training, and test (Figure 3). The washout subset was used as an input to the reservoir with the sole purpose of nullifying the effect of the randomized initial state of the neurons. Then, the training set was fed to the reservoir to collect the data necessary for the aforementioned one-shot training, and the test set was used to evaluate the performance of the ESN.

### 2.4. Experimental Protocol

Figure 4 shows the experimental protocol that was followed to obtain the results. The extraction of the connectome, the filtering of the synapses, and the construction of the connectivity matrix W were performed according to Section 2.1. In accordance with the existing literature [29], the parameter α was set to 0.9, or 1− the timestep, and the input scaling was fixed at 0.7. Note that the *x*, *y*, or *z* time-series are presented as distinct subsets of the ESN neurons, which are selected at random before training with equal probability.

As a result, approximately a third of the ESN neurons receive one of the three Cartesian coordinates of the trajectory. In order to evaluate the *Drosophila* connectome as a reservoir, a comparative study was run with a control architecture through a Monte Carlo simulation. For the control architecture, both the topology and the non-zero weight distribution were generated randomly. For the topology, a connectivity matrix of the desired size was generated, with each element of the matrix having the same probability, corresponding to the desired sparsity of the reservoir, to be initialized as zero. The remaining non-zero elements represent the weight distribution of the reservoir and were generated either using a uniform or Gaussian distribution. Both the connectome-based ESN and the classical implementation were constructed with the same sparsity, reservoir size *N*, input scaling, input matrix $W^{\mathrm{in}}$ and initial output matrix $W^{\mathrm{out}}$ prior to training. Specifically, each neuron in the ESN receives one of the three input time-series, and all neurons are connected to the readout layer; thus, $W^{\mathrm{out}}=\left[1;1;1\right]$, before training. This choice leads to a number of trainable parameters equal to 3×N, thus between 150 for the smallest tested reservoirs and 4500 for the largest ones. The parameters that are taken into consideration for the Monte Carlo simulation are the reservoir size *N*, the selection criteria used to construct the reservoir, the random distribution on the non-zero weights, the β values, and the size of the forecast horizon. Note that the weights of the matrices of both reservoirs are scaled appropriately to ensure that they have the same spectral radius of 0.99, as previously discussed. Thus, the specific absolute weights were determined through this constraint and the imposed distribution. Every combination of the tested hyperparameters was evaluated on five different trajectories, obtained by a 1% variation in the initial conditions, as shown in Figure 3, and randomly re-initialized 10 times, resulting in 50 test cases per hyperparameter combination. In total, four different architectures were compared to each other: the control case, the connectome topology with random weights, the connectome topology with the connectome weights, and the connectome weights with a random topology. This allowed us to isolate the contribution of the connectome topology from that of the synaptic weights. To evaluate the performance of the ESN, the normalized root mean squared error (nRMSE) was computed on both the training and the test set, as follows:(7)$$\mathrm{nRMSE}=\frac{\sqrt{{∥Y-WX∥}^{2}}}{\bar{Y}}$$
where ${∥\cdot ∥}^{2}$ is the squared Euclidean norm and $\bar{Y}$ is the mean of Y.

The Mann–Whitney–Wilcoxon test was used to investigate the statistical significance of the difference in performance of the different architectures [56]. To further characterize the difference in performance, the Bhattacharyya distance was used to define the distance between the nRMSE distributions obtained using two different architectures. The Bhattacharyya distance, $D_{B}$, is a measure of similarity between two probability distributions, often used to quantify the overlap between them, and can be computed as [57](8)$$D_{B}\left(p,q\right)=-\ln\left(\sum_{i}^{n}\sqrt{p_{i}\;q_{i}}\right)$$
where *p* and *q* are datapoints belonging to the nRMSE distributions of the two tested architectures, and *n* is the number of datapoints, 50 in this case. This metric is then averaged across all tested reservoir sizes, *N*, and regularization values, β, in order to estimate the difference in performance between different architectures.

Then, an additional architecture was tested in the same conditions as the other four, keeping both the connectome topology and the connectome weights but shuffling the weights’ values to obtain a random permutation of the original ones. Such an architecture allows us to investigate the synergistic relationship between topology and the distribution of weights in the connectome itself, since it retains both of them but completely changes the mapping between every synapse and its respective weight.

Lastly, a smaller experiment was performed to investigate the role of the spectral radius. The performance of the reservoirs was investigated as a function of the spectral radius in the case of $\beta =10^{-6}$ and a forecast horizon of one timestep for different values of the reservoir size, *N*, selection criteria, and random distributions in the control case. Although previous studies have demonstrated higher performance closer to the edge of chaos, occurring at a spectral radius close to 1, we opted for a more thorough characterization of how the observed performance changes in our specific case for smaller values of the spectral radius.

## 3. Results

### 3.1. Selection Criteria

First, it is important to investigate which subgraph of the entire connectome is taken into consideration and how its characteristics, in terms of the nature of the selected neurons and the topology of the subgraph itself, are affected by the considered size *N*. As mentioned in Section 2.1, we investigated two possible criteria: the $N^{th}$ most connected neurons, either regardless of the classes or proportional to the total class distribution shown in Figure 1. Figure 5 shows the general results regarding the sparsity and the non-zero weight distribution for the selection criteria as a function of *N*.

Starting from the sparsity at N=50, we can observe that the subgraph selected with the ‘most connected’ criterion has a much lower sparsity than the ‘proportional’ criterion, as expected (see Figure 5A). Moreover, the ‘most connected’ criterion leads to monotonically increasing sparsity, which is also likely. As more neurons are added to the reservoir, both selection criteria lead to increasing sparsity, with the ‘most connected’ criterion always leading to a less sparse architecture than the ‘proportional’, as a result of its class-agnostic selection.

Concerning the distribution of extracted weights, Figure 5B shows it as a function of *N*, whereas Figure 5C,D show the probability density function (PDF) of the extracted weights in the case of ‘most connected’ and ‘proportional’ criteria, respectively. Notably, both criteria result in bimodal distributions, with steeper peaks as *N* increases, confirmed by lower values of the variance. However, it is noteworthy that the ‘proportional’ criterion is characterized by the same relative magnitude of the positive and negative peaks in the PDF as the entire connectome, whereas the ‘most connected’ selects more inhibitory than excitatory synapses. Moreover, the ‘proportional’ criterion results in a larger gap in the values both for the entire connectome and for the ‘most connected’ one. This is due to the rescaling factor that is applied to achieve the same spectral radius: for a constant size *N*, a sparser reservoir is rescaled less to achieve the same spectral radius. Conversely, increasing the size *N* generally results in steeper peaks, up to the complete connectome case. According to recent comparisons for connectivity matrix extraction for connectomes [58], the extracted PDF highlights a significant amount of both positive and negative synapses, which is a fundamental requirement for achieving higher learning performance. Moreover, heavy-tailed distributions, like the one extracted from the connectome in this paper, have already been documented as better alternatives for ESNs with respect to the more traditional approaches featuring either Gaussian or uniform distributions [45,59]. However, in our results, the two criteria differ in terms of which peak is prevalent: the ‘most connected’ criterion is characterized by a higher presence of excitatory, or positive, synapses, whereas the ‘proportional’ criterion contains a relatively higher number of inhibitory, or negative, ones.

For a better insight into the selected networks, Figure 6 shows the scatterplots and the class distribution for a few of the selected reservoirs, namely both criteria and N=50,100,700,1500. Compared with the full connectome (see Figure 1), it is obvious that there are major differences. In the case of the ‘most connected’ criterion, N=50 is shown to include a very limited number of classes, with three overwhelmingly represented: ME, LO, and bilateral classes. This trend continues at N=100, but the ME class eventually overcomes the others, representing close to half of the subgraph for N=1500. Moreover, as *N* increases, it is possible to observe a slightly larger number of different classes, but still far from the total number of classes.

Conversely, the ‘proportional’ selection criterion shows a much higher number of different classes already at N=50. However, note that the distribution is not exactly proportional to the one shown in Figure 1 due to the constraint imposed of full connectivity (see Section 2.1). As *N* increases, it can be seen that first ME and later ME>LO start to become more and more represented. Although these two classes are the most represented in the entire connectome, they are still over-represented in the selected subgraph. This inevitably leads to the suppression of the other classes but still maintains a more diverse distribution than the ‘most connected’ criterion.

Finally, the two criteria largely differ in terms of the locations of the selected neurons within the brain. As *N* increases, the ‘most connected’ criterion tends to produce a more symmetrical network, adding neurons almost equally from both hemispheres. In contrast, the ‘proportional’ criterion primarily selects neurons from a single hemisphere, with higher values of *N* corresponding to an even more decentralized subgraph. This can be justified by the criterion trying to include all classes, even the ones that are absent in the more central cluster selected by the previous criterion, thus reaching the further layers. However, the limitation of a fixed *N* prevents the criterion from selecting a sufficient number of sensory neurons symmetrically, thus forcing it to primarily select from a single hemisphere.

As a last remark, the prevalence of ME and ME>LO neurons is hardly surprising. Firstly, the ME neurons represent the second processing stage of the optic lobe, which is known to process visual information, detect edges and motion, and be involved in contrast and color vision. Then, the ME>LO pathway relays information to the higher, more abstract visual areas, contributing to object recognition and image processing. These classes are not only the most abundant in the connectome of an animal that mainly deals with large quantities of visual inputs and processing, but they also represent the middle stages of this processing pipeline, being connected to both the sensory input neurons and the higher processing areas. Thus, these neurons are both the ‘most connected’ and the most over-represented when enforcing the connectedness of the constructed reservoir.

To gain a deeper insight into the topology of the selected reservoir, Figure 7 shows the degree distribution, clustering coefficient distribution, and shortest path distribution, computed as described in Section 2.1. Moreover, it shows such metrics for both selection criteria and N=100 and N=1500, together with a random permutation of the selected reservoirs. Random permutations are obtained by taking the selected reservoir and randomly rearranging the synapses. This allows us to investigate random topologies of the same sparsity and the size of the selected networks.

The results clearly show that the extracted reservoirs have two topological characteristics that drastically differ from their randomized counterparts: the presence of a few highly connected neurons and more degrees of separation between any two given neurons. Random topologies show a much more homogeneous configuration, with low clustering coefficients, a smaller inter-neuron shortest path on average, and a Gaussian degree distribution. Whereas these properties are expected in a random graph [60], they are radically different from what we can observe in the case of the extracted topologies, both for N=100 and N=1500. These topologies have a much higher clustering coefficient, as shown in Figure 7C,D, possibly due to a few super-connected neurons (see Figure 7A,B) and the presence of more separated areas, leading to the results shown in Figure 7E,F. It should also be noticed that such observations are valid for both reported sizes, indicating that the topological properties observed here seem to be intrinsic to any subset of the *Drosophila* connectome.

Notably, the same features can be observed in the topological characteristics of the entire connectome, as shown in Figure 8.

Unlike their random counterparts, all connectome topologies, regardless of their size, contain a much greater number of highly connected clusters. The ‘proportional’ criterion also results in significantly higher clustering coefficients than the ‘most connected’ one, closer to the highest values observed in the complete connectome. Such features have already been observed in several connectomes, including the human connectome [44]. This particular structure, known as small-world organization, has been shown to allow for local specialized processing whilst ensuring efficiency in communication and high robustness, achieved through the presence of multiple clustered fallback routes. Conversely, random topologies are characterized by a more heterogeneous connectivity matrix, with significantly lower clustering coefficients.

### 3.2. Single Simulation

As discussed in Section 2.2, the extracted reservoir is then included in an ESN, which is used for the prediction of multiple CR3BP trajectories. An important step in implementing an ESN is the washout, in which an initial set of points in the time-series is used as an input to the ESN in order to overwrite the initial random state of the neurons. Figure 9A,B show the state of six random neurons during the washout in the case of the classical implementation and the connectome-extracted reservoir, respectively. It can be seen that neurons have a wide range of different behaviors, while the connectome-based architecture displays slightly reduced dynamics, never reaching the boundaries of the state.

To illustrate how these factors can impact the performance of the ESN, Figure 9C–J show the performance of the ESN as a predictor of a single CR3BP trajectory. Note that these results are specific for N=1000, $\beta =10^{-6}$, and a classical implementation obtained with a uniform distribution. Figure 9C,D show that in the case of one timestep prediction, corresponding to 2.7 days, both architectures show good performance, ever so slightly in favor of the connectome-based reservoir. As expected, both architectures see a decrease in performance as the required prediction changes to 3, 7, and then 10 timesteps, which correspond to 8.1, 18.9, and 27 days, respectively. However, the classical implementation shows a more rapid and more drastic deterioration of the performance, with significantly larger loops and greater errors, at least concerning the part of the trajectory around the larger body. However, this single case cannot justify a generalization to the remaining results.

For a more detailed illustration of the origin of the observed difference in performance, Figure 10 shows the same results as in Figure 9 but tested over five different trajectories, with 10 trials per trajectory, and comparing the connectome-based reservoir to a reservoir with non-zero weights sampled from either a uniform or Gaussian distribution, as described in Section 2.4. With this methodology, both architectures are re-initialized for each trial: in the case of the classic implementation, all architecture parameters are randomized, including topology, input connections, synaptic weights, and neuronal biases. Conversely, the connectome-based architecture maintains the same topology and synaptic weights, and only the input connections and biases are randomized. Note that the connectivity matrices of all cases are created with the same sparsity, which is imposed by the topology of the selected connectome subgraph, and they are rescaled to have the same spectral radius, which is restricted to be less than 1 in order to satisfy the ESP, as indicated in the literature [29].

It is clear from the plots that the classic control reservoir tends to overfit the training, achieving a much lower nRMSE when tested on the training set. Conversely, the connectome-based reservoir shows a higher error on the training set but a consistently lower error on the test set. However, both reservoirs show signs of overfitting. This can be explained by the fully connected readout, meaning that all of the reservoir’s neurons are connected to the ridge regression layers. This results in a large number of trainable parameters, which can cause overfitting, especially if paired with lower values of the regularization parameter β.

### 3.3. Monte Carlo Simulations

To fully characterize the connectome-based reservoir both in terms of its topology and the distribution of non-zero weights, a comparative study was performed according to Section 2.4. It is important to note, as previously discussed, that the hyperparameters involved in this investigation are the size of the selected reservoir *N*, the type of distribution used for the selection of random variables, either uniform or Gaussian, the prediction horizon, and the selection criterion used to extract the connectome-based reservoir from the full connectome. All other hyperparameters were kept constant across the tested cases, and every combination was tested 10 times per trajectory over five different trajectories. The four different architectures that are part of this comparative study are as follows:Random topology and random weight distribution, which is the standard way to create ESNs, and represents the control group;Connectome-based topology and extracted weight distribution, utilizing all the data extracted from the *Drosophila* connectome;Connectome-based topology and random weight distribution to solely isolate the contribution of the topology of the connectome;Random topology and extracted weight distribution to solely isolate the contribution of the weight distribution of the connectome.

Figure 11 and Figure 12 show the results of the Monte Carlo simulation for the ‘most connected’ and ‘proportional’ selection criteria, respectively.

Note that the results at $\beta =10^{-6}$ confirm the insights given by the previous observations (see Figure 10), showing a decrease in performance for the classical implementation with respect to the other three. Moreover, it can be seen that such a difference increases as *N* increases. As expected, as the prediction horizon increases, there is a drop in performance (as shown in Figure 9). However, for all tested prediction horizons, the architecture with both a connectome topology and connectome weight distribution does not show a decrease in performance for any tested *N*. In the case of the hybrid architectures, with a connectome topology and random weights or vice versa, the results show that their performance lies between that of the connectome-inspired architecture and the control one. This seems to indicate that both the non-zero weight distribution and the topology play an effective role in counteracting overfitting, given the improvement achieved with just the connectome topology or with its weight distribution. Moreover, the relative performance of the four tested cases is consistent across different conditions, with the random topology and connectome-extracted weights always performing better than the connectome topology and random weights, but worse than the fully biomimetic architecture, for large *N*. This result confirms that both the distribution of weights and the topology play a role in the increase in performance, but the effect of the distribution of weights is stronger.

Next, in the cases with lower regularization (i.e., $\beta =10^{-9}$), it can be observed that the relative performance of all tested architectures is largely unchanged. Nevertheless, even the connectome topology and connectome weight case starts to show overfitting for larger *N*. Noticeably, this phenomenon appears earlier for short prediction horizons (Figure 11A–D and Figure 12A–D) than for longer horizons (Figure 11E–H and Figure 12E–H). This behavior can be explained by the lower regularization, which naturally makes the ESN more prone to overfitting. Moreover, these results indicate that ESNs with a random Gaussian weight distribution are more resilient to overfitting compared to ones initialized with a uniform distribution, with smaller differences in performance between the four tested architectures as *N* increases. Similarly to the previous case, the distribution of weights taken from the connectome shows a stronger effect on the error reduction than the topology when considered in isolation, and the fully biomimetic architecture, despite starting to show a decrease in performance as *N* increases, is still the most resilient.

Conversely, the ESN behavior at $\beta =10^{-3}$ shows drastically different trends. On one hand, there is no decrease in performance as *N*, and thus the number of trained degrees of freedom, increases. This result reinforces the conclusion that overfitting is the origin of the decrease in performance since higher β values specifically counteract overfitting. On the other hand, the connectome topology and connectome weight architecture shows the highest error almost consistently in different conditions. However, the connectome topology with random weight initialization still closely matches the performance of the classical implementation, and larger reservoirs with a uniform weight distribution eventually outperform the control architecture with random topology and random weights.

Comparing the corresponding subfigures in Figure 11 and Figure 12, the results are extremely similar, despite the differences in the topology of the selected connectome subgraphs that constitute the reservoirs (see Section 2.1). The most noticeable difference between the two Monte Carlo simulations is that the results corresponding to the architecture with the connectome topology and random weights are closer to those for the control architecture in the case of ‘proportional’ selection. Moreover, this combination of topology and weight distribution is closer in performance to the control case than the architecture with connectome weights and a random topology, which in turn is closer to the complete connectome-inspired architecture. This indicates that the contribution of the topology towards the mitigation of overfitting is present but small, and the observed bimodal distribution of non-zero weights (see Figure 5) plays a bigger role.

To further investigate the synergistic roles of synaptic weights and topology, the case of a connectome topology and shuffled connectome weights was also tested. As discussed in Section 2.4, this particular architecture has both the same non-zero weight distribution and the same topology as the fully biomimetic connectome-inspired one but does not have the same mapping between the synapses and their respective weights. The performance of this architecture is statistically equivalent to the previously tested case of connectome weights and a random topology, as the Mann–Whitney–Wilcoxon test shows a *p*-value > 0.05 for all tested combinations of *N* and β. Moreover, the Bhattacharyya distance between the nMRSE distribution of the two architectures was computed and averaged over all reservoir sizes, *N*, and all regularization values, β, resulting in $3.2\times 10^{-3}\pm 2\times 10^{-3}$ and $1.6\times 10^{-3}\pm 1\times 10^{-3}$ in the case of the ‘most connected’ and the ‘proportional’ selection criterion, respectively. These values indicate that shuffling the weights extracted from the connectome has the same effect on the performance as randomizing the topology of the reservoir. This indicates not only that both the connectome weight distribution and its topology play a role in the observed overfitting resilience but also that they are strongly bound to each other. In the case of the fully biomimetic connectome-inspired design, the position of every synaptic weight within the topology is key to achieving the observed properties. While this was outside the scope of this study due to the sheer size of the networks being explored, one potential avenue for future study would be to investigate how the performance of the ESN declines as progressively larger portions of the weights are shuffled.

As discussed in Section 2.1, a spectral radius of 0.99 was selected in accordance with previous literature on ESNs and biomimetic ESNs. However, in order to further characterize the difference in performance between the connectome-based reservoir and the control architecture, a set of simulations with lower spectral radii, such as 0.75, 0.5, and 0.25, were performed. These simulations follow the same experimental protocol as the previous ones, but they only feature reservoirs with regularization $\beta =10^{-6}$ and a forecast horizon of one timestep. Figure 13 showcases the difference in nRMSE between the control case and connectome-based reservoirs, respectively, as a function of reservoir size *N* for different selection criteria and random distributions.

It is noteworthy that, in agreement with previous studies [44,45], connectome-based architectures perform better at high spectral radii. Among the tested cases, while the biomimetic reservoirs always outperform the classical ones for a spectral radius of 0.99, they are equivalent, and even underperform, for lower ones, especially for smaller sizes. In the case of a spectral radius of 0.25, the classical ESN consistently performs better than the connectome-based ESN, with very few exceptions. In summary, a spectral radius of 0.99 yields better performance than at least 0.25, 0.5, and 0.75 in both models.

All investigated conditions seem to lead to the conclusion that both synaptic weights and the reservoir topology play a role, albeit of different magnitude, in preventing overfitting as *N* increases within a certain range of the regularization parameter, β. This phenomenon can be overshadowed by large β values, and its effectiveness seems to be lower for very low values of β. However, the effect of $\beta <10^{-9}$ has not been investigated as the performance of the ESNs at $\beta =10^{-9}$ already shows high nRMSE for relatively small *N*.

### 3.4. Entire Connectome Simulations

The results investigated up to this section were all obtained using a relatively small reservoir, up to N=1500. Whilst the tested sizes are coherent with state-of-the-art reservoir computing, they are still several orders of magnitude smaller than the size of the actual brain. Since the purpose of this work is to investigate the connectome as a reservoir, a final study using the entire connectome was performed. As mentioned in Section 2.1, the entire connectome, after quality filtering, consists of 104,909 neurons, forming a single connected graph. Therefore, it is not computationally possible to keep all neurons connected to the readout layer, namely $W^{\mathrm{out}}=\left[1;1;1\right]$, as the single-shot training needed by ridge regression (see Equation (Equation 6)) relies on the inversion of the matrix $XX^{T}+\beta I$, which is an n×n matrix, where *n* is the number of non-zero elements in $W^{\mathrm{out}}$. As a solution, we limited both the neurons connected to the output and the ones receiving the data from the input according to Table 2 and based on the *Drosophila*’s neurophysiology.

Neurons belonging to the input classes receive the input signal, whereas the ones belonging to an output class are connected to the readout layer. The proposed rough classification is loosely based on the *Drosophila*’s neurophysiology [61], and it is used solely to reduce the computational complexity of the training process. Moreover, only for this part of the investigation, each trajectory was tested 3 times instead of 10 to compensate for the large computational time required due to the dimensionality of the matrices involved. Similarly, only a single-timestep prediction was attempted. Figure 14 shows a comparison between ESN architectures with a random topology and a random weight distribution on one hand and a connectome-based topology and an extracted weight distribution on the other.

The results confirm the trends that were already observed in the Monte Carlo simulations. On one hand, when $\beta =10^{-3}$, it can be seen that the control architecture outperforms the connectome-based one on both the training and the test sets. On the other hand, the results for $\beta =10^{-6}$ are drastically different, demonstrating the connectome-based architecture’s resilience to overfitting. The connectome-based architecture achieves this despite the large number of trainable parameters, just over $22\times 10^{3}$, three times the number of all neurons belonging to the output classes, which increases the chance of overfitting. An interesting result is that the connectome-based architecture with $\beta =10^{-6}$ not only outperforms the control case for the same β but also outperforms both architectures when $\beta =10^{-3}$, with an nRMSE <2%. Once more, the connectome-based reservoir shows much higher resilience to overfitting, as indicated by the small difference in performance between the training and test phases.

## 4. Conclusions

In this work, the possibility of using the scanned connectome of a *Drosophila* as a reservoir for an ESN is investigated, and its performance as a time-series predictor is characterized through a comparative study. As per state-of-the-art guidelines [29], the proposed and control architectures are compared while enforcing the same spectral radius, reservoir size, number of neurons that receive the input, input scaling, and number of neurons that are connected to the readout layer. The two main features of the architecture that are investigated are the topology of the reservoir itself and the distribution of non-zero weights. In the control case, the topology is randomized, and the non-zero weight distribution is either Gaussian or uniform. To further isolate the role of the topology, hybrid architectures are also added to the study: one with the topology based on the connectome and a Gaussian or uniform distribution of non-zero weights and one with a randomized topology and the weights that have been extracted from the connectome.

In order to consider reservoir sizes similar to those used in previous literature, two possible selection criteria are adopted. Noticeably, the two criteria lead to different neurons being selected to participate in the connectome-based ESN, where the neurons differ in terms of their biological class and their position within the brain. Although the ‘most connected’ criterion results in a more symmetric selection while the ‘proportional’ criterion primarily selects neurons from the right hemisphere, the results show that the performance is not affected by the selection criterion. However, regardless of the selection criterion, all connectome-based reservoir topologies show a high degree of clustering. We observe, in line with previous literature, that highly connected neurons provide robustness, and shortcuts between anatomically distant brain regions ensure fast communication. Thus, we hypothesize that the increased robustness of such topologies with respect to the more heterogeneous ones of the control cases can lead to the observed higher resilience to overfitting. Concerning the extracted synaptic weights, the reservoirs are characterized by a heavy-tailed bimodal distribution. This is also consistent with previous studies on both traditional and connectome-inspired ESNs and has been shown to lead to increased learning performance with respect to other more traditional distributions, such as uniform or Gaussian. All in all, our findings support a possible generalization of the results of this study: despite working on significantly smaller reservoirs than the size of the entire brain, the results are consistent across different connectome regions, indicating that the investigated properties are not localized but rather extend across the connectome.

Regarding the main results of the study, Monte Carlo simulations are run to span over a wide range of parameters, including the regularization of the readout layer, the considered ESN size, the size of the prediction horizon, and the random distribution governing the control architecture. The outcome indicates that the connectome-based reservoir is significantly more resilient to overfitting than a standard implementation. Weaker readout normalization, which potentially allows for more pronounced overfitting, was found to have a much smaller effect on the connectome-based ESN than the control one. Specifically, the fully biomimetic ESN, with both the topology and weights extracted from the connectome, was significantly less prone to overfitting with increasing reservoir size compared to the other studied architectures. As previously discussed, we believe that the increased robustness due to the higher clustering and the small-world topology, particularly in the case of the ‘proportional’ selection criterion, and the higher learning ability of bimodal heavy-tailed distributions for synaptic weights are the main factors behind the observed behavior. The case in which the connectome-based reservoir shows the worst performance is for high values of the regularization parameter, in which case overfitting is much less likely to happen in the first place.

Additionally, the performance of the ESN with a connectome topology and random non-zero weights is almost always within the range defined by the performance of the other two architectures, indicating that both the topology and extracted weights play a role in the observed resilience to overfitting. Notably, our investigation revealed that the unique mapping between the weight distribution and the topology of the connectome is key to further augmenting this property, since a permutation of the synaptic weights results in a performance that is statistically identical to that of a connectome with a completely random topology and weights. Thus, anatomical features, such as the small-world organization of the topology of the reservoir, and physiological ones, such as the distribution of synaptic weights, cannot be treated independently. In other words, both the presence of clusters and the relative weights of all redundant pathways within them are crucial to achieving a higher resilience to overfitting. The exception is the case of high regularization, $\beta =10^{-3}$, in which the hybrid architecture occasionally shows the best performance for higher reservoir sizes, such as 1300 and 1500.

An investigation was also conducted using the entire connectome as a reservoir. Given the large number of trainable parameters, overfitting is to be expected, and such is the case for the control architecture. Conversely, the connectome-based architecture is characterized by minimal overfitting, with similar errors on the training and test sets. Moreover, the connectome-based ESN shows a slightly better performance with lower regularization, with an error at $\beta =10^{-6}$ that is also lower than that of both tested architectures at $\beta =10^{-3}$. This further supports the link between the properties of the reservoir and its resilience to overfitting, especially for larger reservoirs.

This study is an attempt to understand the computing capabilities of biological brains within the framework of time-series prediction enabled by AI and neural networks. We acknowledge the limitations of the investigation regarding the tested conditions, application tasks, and the size of the study itself. Thus, we encourage future work aimed at the expansion and generalization of the results hereby showcased, starting from the extension of the complete connectome as a reservoir beyond the specific task of time-series prediction. Additional future work can focus on alternative biologically plausible selection criteria in order to unveil additional properties that might be strongly related to a particular subsection of the connectome. Lastly, a study on other animals’ full connectomes, as soon as complete scans are available, is clearly the key to understanding if the observed resilience to overfitting is a property of biological brains in general, as we are hypothesizing, or if it is specifically a property of the brain of *Drosophila*. Lastly, a possible future research direction could be the physical implementation of biomimetic connectome-based reservoirs, ranging from more abstract artificial implementations of neuromorphic brain-like networks, such as clustered nanowires [62,63], to the more drastic approach of using real biological brains or neurons, exploiting organoid intelligence [64].

In summary, this study has demonstrated that, in the framework of reservoir computing for time-series prediction, both the topology and the synaptic weights of a real biological brain have an edge with respect to state-of-the-art implementation with respect to overfitting. Such results unveil a peculiar property of the brains of *Drosophila* and potentially other animals. We conjecture that, considering the complex nature of animals and the vast range of tasks that their bodies need to accomplish at any given time, overfitting to a specific task would lead to a drop in other functions.

## Figures and Tables

**Figure 1 biomimetics-10-00341-f001:**
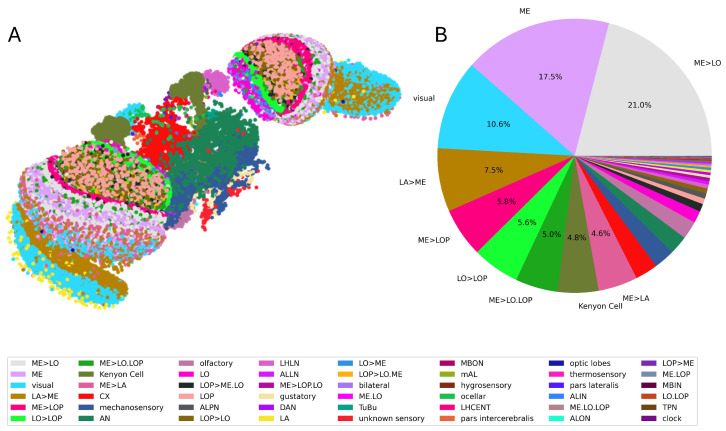
(**A**) A 3D scatterplot of the entire connectome and (**B**) a pie chart of the class division within the connectome itself.

**Figure 2 biomimetics-10-00341-f002:**
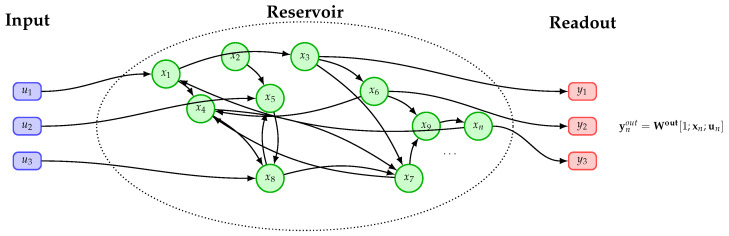
Diagram of an ESN in which a subset of neurons in the reservoir (green) receive the input signal from the input neurons (blue). The readout layer (red), which is usually a fully connected linear layer, extracts activations from another subset of neurons and produces the output.

**Figure 3 biomimetics-10-00341-f003:**
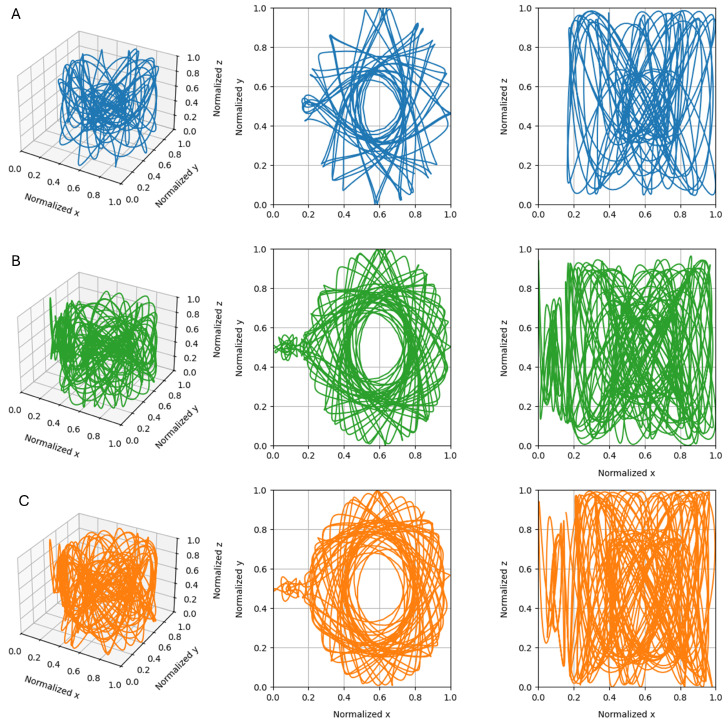
(**A**) Washout, (**B**) train, and (**C**) test subsets of the CR3BP trajectory considering r=[−0.80,0.0,0.0] and v=[0,−0.63,0.08].

**Figure 4 biomimetics-10-00341-f004:**
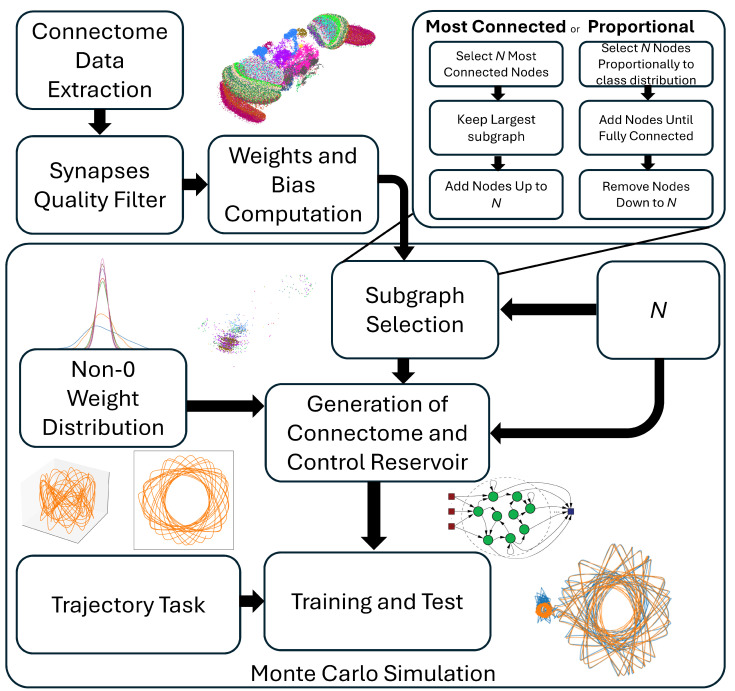
Flow chart of the experimental protocol. The raw data are extracted from the database and filtered, and the connectivity matrix W of the connectome is computed. Then, one of two possible selection criteria is used to extract a reservoir of size *N*, and both the connectome-inspired and the control reservoir are generated, with the same size, same input scaling, same $W^{\mathrm{in}}$, and same initial $W^{\mathrm{out}}$, prior to training. To evaluate the ESN, five trajectories are created and used to train and test the generated ESNs. A grid search is run over the values of *N*, β values, the random distribution of the non-zero weights, and however many timesteps into the future the ESN is trying to predict.

**Figure 5 biomimetics-10-00341-f005:**
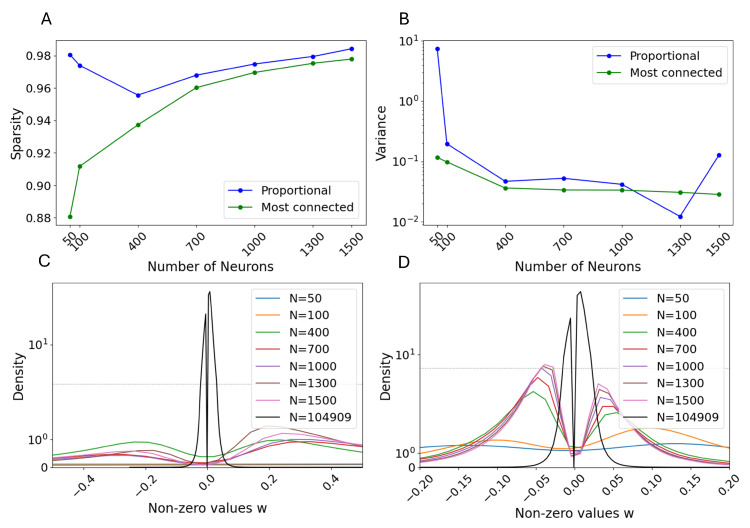
(**A**) Sparsity and (**B**) variance of the non-zero weights as a function of the number of neurons selected in the reservoir. The probability density function of the extracted weight in the case of (**C**) the ‘proportional’ criterion and (**D**) the ‘most connected’ criterion. For (**C**,**D**), the dotted gray line represents the threshold between the linear *y* scale, below the line, and the logarithmic *y* scale, above.

**Figure 6 biomimetics-10-00341-f006:**
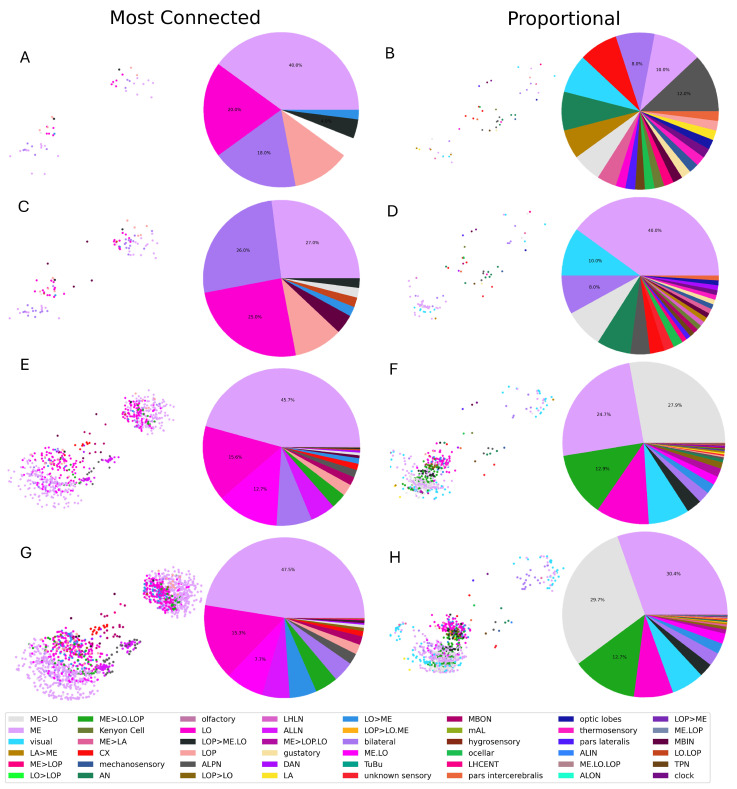
A 3D scatterplot of the selected reservoir and a pie chart of the class division within such a reservoir in the case of (**A**) the ‘most connected’ criterion and N=50, (**B**) the ‘proportional’ criterion and N=50, (**C**) the ‘most connected’ criterion and N=100, (**D**) the ‘proportional’ criterion and N=100, (**E**) the ‘most connected’ criterion and N=700, (**F**) the ‘proportional’ criterion and N=700, (**G**) the ‘most connected’ criterion and N=1500, and (**H**) the ‘proportional’ criterion and N=1500.

**Figure 7 biomimetics-10-00341-f007:**
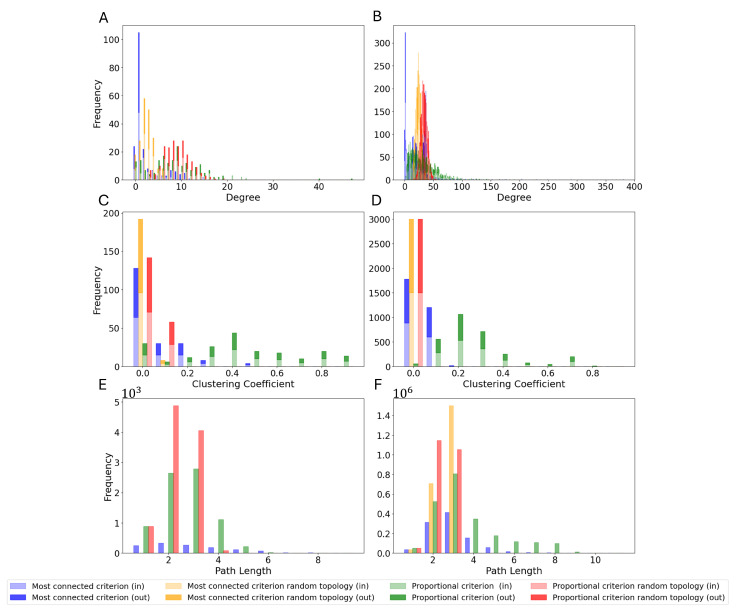
Topology metrics of the selected reservoirs: (**A**) degree distribution for N=100, (**B**) degree distribution for N=1500, (**C**) clustering coefficient distribution for N=100, (**D**) clustering coefficient distribution for N=1500, (**E**) shortest path distribution for N=100, (**F**) shortest path distribution for N=1500. Note that for the shortest path, directionality is considered while computing the path and therefore not indicated with different color intensities.

**Figure 8 biomimetics-10-00341-f008:**
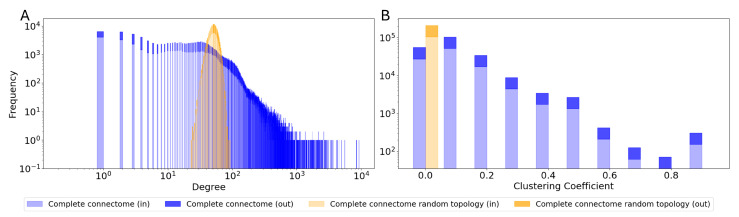
Topology metrics of the entire connectome as a reservoir: (**A**) degree distribution and (**B**) clustering coefficient distribution.

**Figure 9 biomimetics-10-00341-f009:**
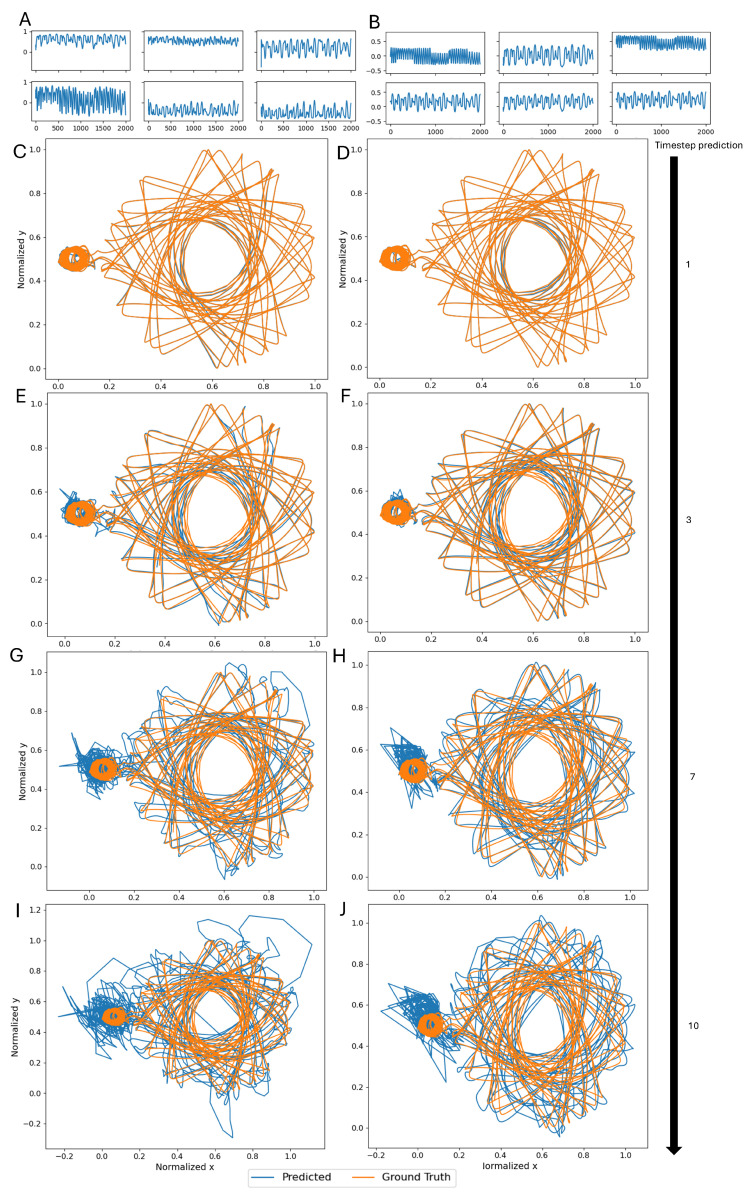
The neuronal activation state of six random neurons during the washout in the case of (**A**) a classical reservoir with a uniform non-zero weight distribution and (**B**) the reservoir extracted from the connectome. An example of the performance of the ESN on a single test trajectory in the case of N=1000, $\beta =10^{-6}$, and a classical implementation obtained with a uniform distribution in the case of (**C**) a classical and (**D**) connectome-based reservoir and a 1-timestep prediction; (**E**) a classical and (**F**) connectome-based reservoir and a 3-timestep prediction; (**G**) a classical and (**H**) connectome-based reservoir and a 7-timestep prediction; and (**I**) a classical and (**J**) connectome-based reservoir and a 10-timestep prediction.

**Figure 10 biomimetics-10-00341-f010:**
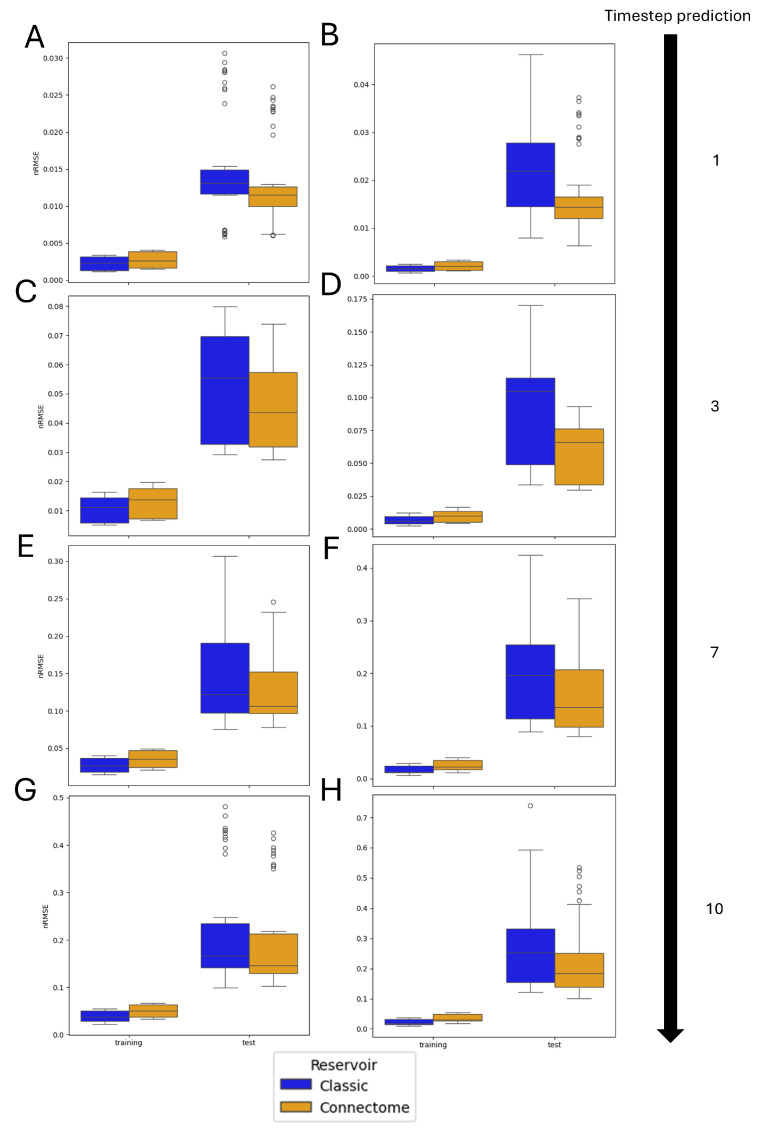
Comparative results of the ESN performance of the training and test sets in the case of N=1000 and $\beta =10^{-6}$. Connectome and classic refer to the connectome-based and random reservoirs, respectively. The selection criterion used for the extraction of neurons for the connectome-based ESN is the ‘most connected’ criterion. Simulations were performed for the following cases: (**A**) Gaussian and (**B**) uniform non-zero weight distribution for the classic reservoir in the case of 1-timestep prediction, (**C**) Gaussian and (**D**) uniform non-zero weight distribution for the classic reservoir in the case of 3-timestep prediction, (**E**) Gaussian and (**F**) uniform non-zero weight distribution for the classic reservoir in the case of 7-timestep prediction, and (**G**) Gaussian and (**H**) uniform non-zero weight distribution for the classic reservoir in the case of 10-timestep prediction.

**Figure 11 biomimetics-10-00341-f011:**
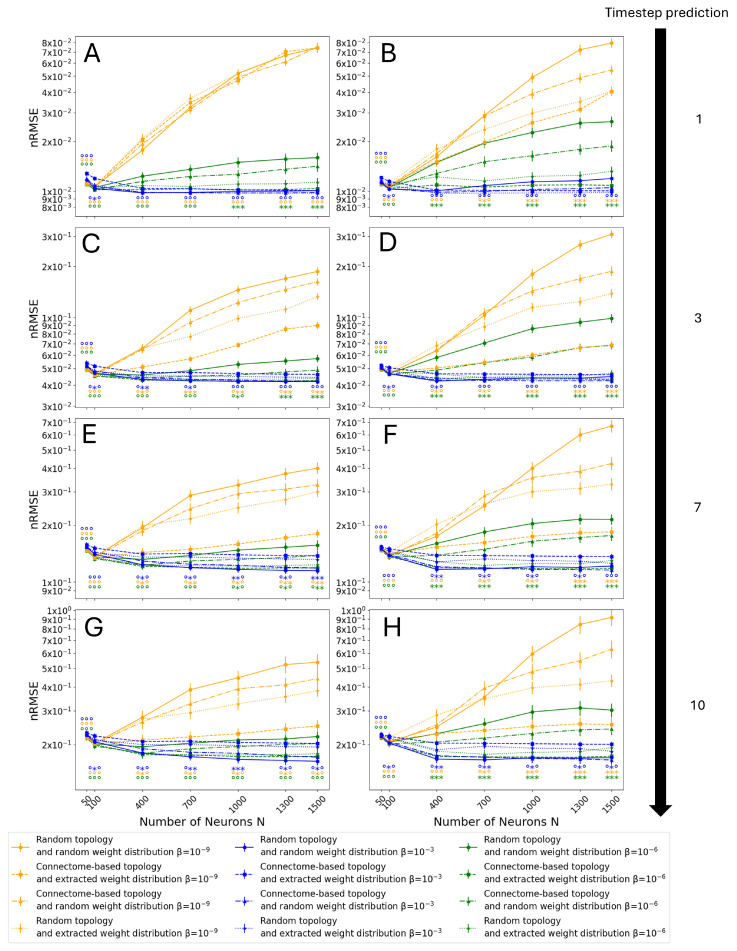
Results on the test set as a function of reservoir size, *N*, and normalization factor, β, using the ‘most connected’ selection criterion. Simulations were performed for the following cases: (**A**) Gaussian and (**B**) uniform non-zero weight distribution for the classic reservoir with 1-timestep prediction, (**C**) Gaussian and (**D**) uniform non-zero weight distribution for the classic reservoir with 3-timestep prediction, (**E**) Gaussian and (**F**) uniform non-zero weight distribution for the classic reservoir with 7-timestep prediction, and (**G**) Gaussian and (**H**) uniform non-zero weight distribution for the classic reservoir with 10-timestep prediction. Error bars display the mean’s standard deviation. Statistically significant differences (p<0.05) under the Mann–Whitney–Wilcoxon test are indicated by the marker ∗, and no significance is indicated by the marker ∘ as follows: the control case of random topology and random weight distribution was tested against the connectome-based topology and random weight distribution (left marker), the connectome-based topology and extracted weight distribution (central marker), and the random topology and extracted weight distribution (right marker).

**Figure 12 biomimetics-10-00341-f012:**
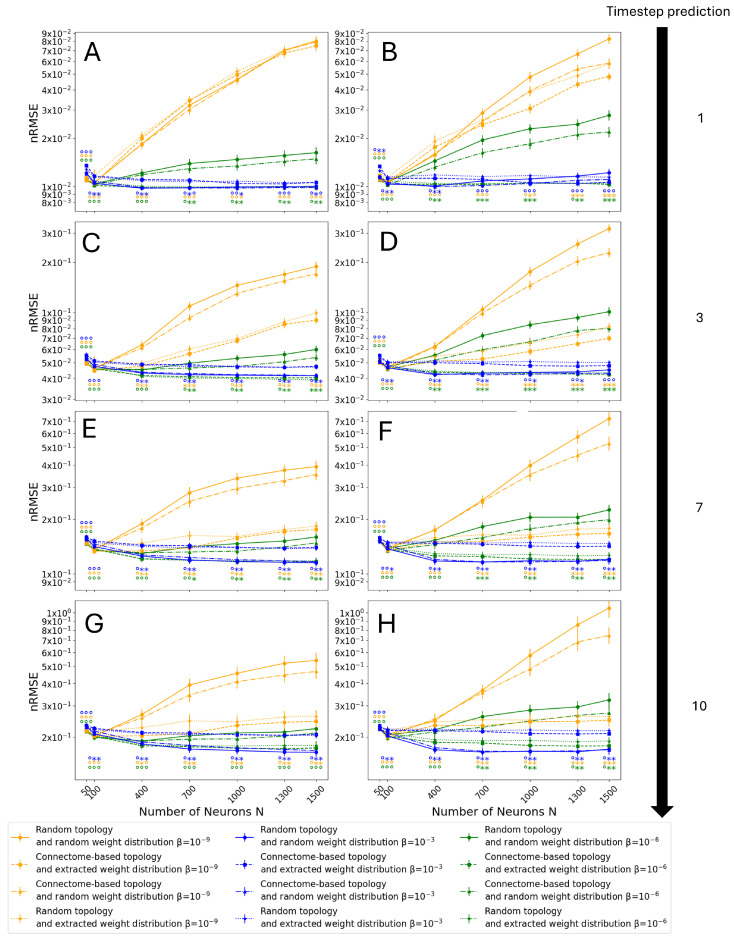
Results on the test set as a function of reservoir size, *N*, and normalization factor, β, using the ‘proportional’ selection criterion. Simulations were performed for the following cases: (**A**) Gaussian and (**B**) uniform non-zero weight distribution for the classic reservoir with 1-timestep prediction, (**C**) Gaussian and (**D**) uniform non-zero weight distribution for the classic reservoir with 3-timestep prediction, (**E**) Gaussian and (**F**) uniform non-zero weight distribution for the classic reservoir with 7-timestep prediction, and (**G**) Gaussian and (**H**) uniform non-zero weight distribution for the classic reservoir with 10-timestep prediction. Error bars display the mean’s standard deviation. Statistically significant differences (p<0.05) under the Mann–Whitney–Wilcoxon test are indicated by the marker ∗, and no significance is indicated by the marker ∘ as follows: the control case of a random topology and random weight distribution was tested against the connectome-based topology and random weight distribution (left marker), the connectome-based topology and extracted weight distribution (central marker), and the random topology and extracted weight distribution (right marker).

**Figure 13 biomimetics-10-00341-f013:**
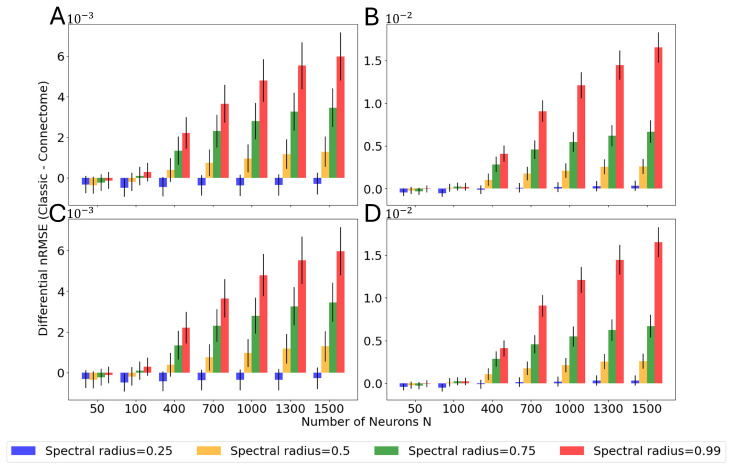
Differential nRMSE between classical and connectome nRMSE, respectively, for $\beta =10^{-6}$ and one-timestep forecast horizon as a function of reservoir size, *N*, in the case of (**A**) a Gaussian and (**B**) uniform non-zero weight distribution for the classic reservoir and ‘most connected’ selection criterion for the connectome-based reservoir and (**C**) a Gaussian and (**D**) uniform non-zero weight distribution for the classic reservoir and ‘proportional’ selection criterion for the connectome-based reservoir.

**Figure 14 biomimetics-10-00341-f014:**
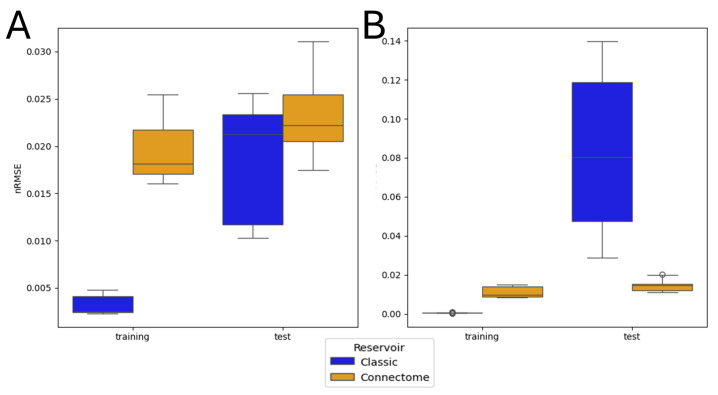
Comparative results of the ESN performance of the training and test sets in the case of the whole connectome predicting one timestep into the future. The control architecture uses a uniform distribution of non-zero weights. Connectome and classic refer to the connectome-based and manually constructed reservoirs, respectively. This study was performed in the following conditions: (**A**) $\beta =10^{-3}$ and (**B**) $\beta =10^{-6}$.

**Table 1 biomimetics-10-00341-t001:** Neuronal class abbreviation, complete name, and brief function description for all classes present in the connectome.

Neuron Class	Full Name	Function	Neuron Class	Full Name	Function
CX	Central Complex	Motor control, navigation	Kenyon Cell	Kenyon Cell	Learning and memory
ALPN	Antennal Lobe Projection Neuron	Olfactory signal relay	LO	Lamina Output Neuron	Early visual processing
bilateral	Bilateral Neuron	Cross-hemisphere connections	ME	Medulla Neuron	Visual processing
ME>LOP	Medulla to Lobula Plate	Motion detection	ME>LO	Medulla to Lobula	Higher-order vision
LO>LOP	Lobula to Lobula Plate	Object motion detection	ME>LA	Medulla to Lamina	Visual contrast enhancement
LA>ME	Lamina to Medulla	Photoreceptor signal relay	ME>LO.LOP	Medulla to Lobula and Lobula Plate	Motion and feature integration
ALLN	Allatotropinergic Neuron	Hormonal regulation	olfactory	Olfactory Neuron	Detects airborne stimuli
DAN	Dopaminergic Neuron	Learning, reward, motivation	MBON	Mushroom Body Output Neuron	Readout of learned behaviors
ME>LOP.LO	Medulla to Lobula Plate and Lobula	Visual integration	LOP>ME.LO	Lobula Plate to Medulla and Lobula	Motion-sensitive feedback
LOP	Lobula Plate Neuron	Optic flow detection	LOP>LO.ME	Lobula Plate to Lobula and Medulla	Visual-motor integration
LOP>LO	Lobula Plate to Lobula	Motion-sensitive projection	LA	Lamina Neuron	First visual synaptic layer
AN	Antennal Neuron	Mechanosensory and olfactory processing	visual	Visual Neuron	General vision processing
TuBu	Tubercle Bulb Neuron	Connects brain to vision	LHCENT	Lateral Horn Centroid	Odor valence processing
ALIN	Antennal Lobe Interneuron	Olfactory modulation	mAL	Mushroom Body-associated Antennal Lobe Neuron	Olfactory-learning link
LHLN	Lateral Horn Local Neuron	Innate odor-driven behavior	ME.LO	Medulla and Lobula Neuron	General vision processing
mechanosensory	Mechanosensory Neuron	Touch, vibration detection	ME.LO.LOP	Medulla, Lobula, and Lobula Plate Neuron	Multi-layer vision processing
LO>ME	Lobula to Medulla	Visual feedback	hygrosensory	Hygrosensory Neuron	Detects humidity
pars lateralis	Pars Lateralis Neuron	Hormonal/circadian regulation	unknown sensory	Unknown Sensory Neuron	Uncharacterized sensory function
LO.LOP	Lobula and Lobula Plate Neuron	Motion and space awareness	ocellar	Ocellar Neuron	Light intensity detection
optic lobes	Optic Lobe Neuron	General vision processing	pars intercerebralis	Pars Intercerebralis Neuron	Hormonal regulation
ME.LOP	Medulla and Lobula Plate Neuron	Motion and edge detection	gustatory	Gustatory Neuron	Taste processing
ALON	Antennal Lobe Olfactory Neuron	Olfactory cue processing	MBIN	Mushroom Body Input Neuron	Learning circuit modulation
thermosensory	Thermosensory Neuron	Temperature sensing	clock	Clock Neuron	Circadian rhythm control
LOP>ME	Lobula Plate to Medulla	Motion-sensitive feedback	TPN	Transmedullary Projection Neuron	Optic lobes to brain

**Table 2 biomimetics-10-00341-t002:** Classification of neuron classes into input, intermediate, and output categories. The input neurons receive the input signal, whereas the output neurons are connected to the readout.

Type	Neuron Classes
Input	olfactory, visual, mechanosensory, hygrosensory, thermosensory, gustatory, ocellar, unknown sensory
Intermediate	CX, ALPN, LO, bilateral, ME, ME>LOP, ME>LO, LO>LOP, ME>LA, LA>ME, ME>LO.LOP, ALLN, LOP>ME.LO, LOP>LO.ME, LA, AN, ALIN, mAL, LHLN, ME.LO, ME.LO.LOP, LO>ME, optic lobes, ME.LOP, TPN
Output	MBON, DAN, LHCENT, clock, pars intercerebralis, pars lateralis, Kenyon Cell, ALON, LOP>ME, LOP>LO.ME, LOP>LO, LOP, TuBu

## Data Availability

The source code used to run the simulations and obtained results can be found at https://gitlab.com/EuropeanSpaceAgency/fly_connectome, accessed on 20 May 2025.

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
