# Peer review of "The Drosophila Connectome as a Computational Reservoir for Time-Series Prediction"

_biomimetics, 2025, doi:10.3390/biomimetics10050341_

Round 1

Reviewer 1 Report

Comments and Suggestions for Authors

This study constructs a reservoir computing model based on the detailed connectome of Drosophila and investigates its learning capabilities. The authors assign synaptic weights using the synapse size (referred to as the connection score), with the sign of each weight determined by the type of neuronal cell. To reduce the size of the reservoir, they extract a subgraph by selecting either (1) the top N neurons with the highest degree from the full set of 104,909 neurons, or (2) the top N high-degree neurons while maintaining the original distribution of neuronal cell types. The reservoir model is evaluated on a trajectory prediction task involving P-step-ahead forecasting in the three-body problem. The connectome-based reservoir demonstrates superior prediction performance compared to a reservoir constructed with a random weight matrix. This is an intriguing study that explores how the structural properties of a biological connectome relate to computational and learning performance. However, the experimental setup for comparison is not clearly described, and the handling of the spectral radius—a critical parameter in reservoir computing—is insufficiently addressed. Additionally, the originality of the study in relation to prior work should be more explicitly highlighted. Major comments are as follows:

(1) Previous studies have constructed recurrent neural networks based on the connectome to investigate their performance in reservoir computing, including work using the human brain connectome [1, 2]. These studies should be cited in the introduction to properly contextualize the current work. Furthermore, as acknowledged by the authors, a previous study [3] employing the Drosophila connectome reported improved performance in a time-series prediction task using a subset of the connectome. The distinction between that study and the present work must be clearly articulated to highlight the originality of the current research. If the only difference lies in the use of the full connectome versus a partial one, the contribution may be perceived as incremental. The authors are encouraged to explicitly state the key methodological advances or novel findings introduced in this study.

(2) Figure 7 presents three topological indices. Do these metrics account for synaptic weights? Based on equation (1), they appear to be defined for undirected, unweighted (binary) graphs. To more accurately characterize the structure of the connectome, topological indices for weighted, directed graphs should be employed. Additionally, it is important to verify that the subgraph extracted for the reservoir preserves the topological properties of the original network. To this end, the authors should analyze the topological indices of the full connectome (comprising 104,909 nodes) and compare their distributions with those of the selected subgraph.

(3) Section 2.4 lacks clarity regarding the procedure used to generate control reservoirs for comparison. What exactly does the term “randomized” refer to in this context? Were the synaptic weights assigned random values, or were existing weights shuffled? Figures 10 and 11 refer to “random topology” and “random weight distribution,” whereas Figure 7 uses the term “shuffled,” leading to potential confusion. The authors should unify and clarify their terminology to avoid ambiguity and ensure methodological transparency.

(4) Regarding point (3), if the “classic reservoir” refers to a model with randomly assigned weights, then results for the condition using shuffled weights should also be presented. Previous studies [1, 2] have compared original and shuffled weight configurations to isolate the effect of network topology from that of weight distribution. Including this condition would enable a more rigorous evaluation of the contribution of topological features to learning performance while controlling for the distribution of synaptic weights.

(5) The treatment of the spectral radius—a critical parameter in reservoir computing—is insufficiently detailed. While the manuscript states that spectral radii were unified to a common value, neither the specific value nor the normalization procedure for the weight matrices is provided. Prior work [1, 2] has demonstrated that the topological features of the connectome, particularly small-worldness or clustering organizations, can modulate the sensitivity of learning performance to the spectral radius (i.e., the optimal spectral radius value may shift depending on topology). Therefore, directly comparing reservoirs with different topologies but identical spectral radii may not yield fair or informative conclusions. To address this, the authors should present performance curves across a range of spectral radii to determine whether the connectome-based reservoir outperforms others consistently, or only within a narrow parameter range.

(6) The fact that only a single task—time-series prediction—was used to evaluate the reservoir is a clear limitation of the study. In contrast, previous studies [1, 2] employed multiple benchmark tasks, including memory capacity evaluation, which is widely recognized as a standard test in reservoir computing research. Incorporating an additional task would significantly strengthen the validity and generalizability of the study’s conclusions.

(7) The manuscript does not address the underlying reasons for the improved predictive performance observed in the connectome-based reservoir. Prior work [1] has attributed such improvements to specific topological features, such as small-world organization. A discussion, even speculative, regarding the mechanisms that might account for the observed performance gains—supported by existing findings—would provide valuable insight and enrich the contribution of the study.

[1] Kawai et al., “A small-world topology enhances the echo state property and signal propagation in reservoir computing,” Neural Networks, 112, 15-23, 2019.

[2] Suarez et al., “Learning function from structure in neuromorphic networks,” Nature Machine Intelligence, 3, 771-786, 2021.

[3] Morra & Delay, “Imposing connectome-derived topology on an echo state network,” IJCNN, 2022.

Author Response

Please find our reply in the attached document.

Reviewer 2 Report

Comments and Suggestions for Authors

This study takes the fruit fly connectome as a blueprint to reasonably construct a reservoir structure, demonstrating stronger anti-overfitting capabilities compared to standard implementations. The research is exploratory, meaningful, and insightful.

Comments about biomimetics-3613448

Reservoir Computing (RC) is an efficient training framework for recurrent neural networks (RNNs) that has garnered significant attention in the fields of machine learning and dynamical systems modeling. The optimization and implementation of reservoir architectures lie at its core. Among these, bio-inspired reservoirs represent a critical research direction. Beyond employing spiking neurons (e.g., the Leaky Integrate-and-Fire (LIF) model) to construct spiking neural networks (SNNs) that emulate biological temporal coding, another approach involves leveraging cultured living neuronal networks to develop bio-inspired reservoirs.

Using cultured living neuronal networks as physical substrates for reservoir computing seems to return to the very essence of brain-inspired computing. However, this approach presents substantial challenges, particularly in compatibility with today's highly advanced microelectronic fabrication technologies. Alternatively, mapping the connectivity of connectomes from different species directly onto reservoirs—enabling biological mimicry—emerges as a promising solution that combines the strengths of multiple approaches.

However, that study was limited both by the limited range of investigated hyperparameters and by the size of the investigated connectome, as at the time there was no reported case of a complete scan. In particular, whereas the reported result of connectome-based topology and randomized weights seems to imply a performance at best comparable to the standard fully randomized reservoir, the attempt at simulating also the synaptic weights was unsuccessful, further lowering performance for the heterogeneous connectivity case and completely failing for the homogeneous one. Following that study, connectome topologies have been proposed as an additional constraint on top of standard practice”, cited from original manuscript.

Fortunately, the complete connectome of a fruit fly brain has been recently published in Nature 2024, In this work, the possibility of using the scanned connectome of a Drosophila as a reservoir for an ESN is investigated, and its performance is characterized through a comparative study. Cited from original manuscript.

This is a complex undertaking that employs unique methodologies in mapping the fruit fly connectome to the reservoir, with many parameter selections remaining open for discussion. The work is inherently exploratory in nature and will inevitably leave numerous questions for future investigation. I choose not to interrogate each one individually.

Author Response

(The authors gave the same response as above.)

Reviewer 3 Report

Comments and Suggestions for Authors

The presented work investigates the possibility of using connectome of a Drosophila as a reservoir in a reservoir computing system (RCS). The performance of such RCS is characterized through a comparative study. Two types of simulation approaches were used and the results were compared and analysed. Overall, the presented results are sound and promising, so the manuscript could be published after minor revision:

  1. Commonly the dataset is split into three sub datasets: training, validation and testing. In the presented work the validation dataset is not used, however this could affect e.g. the performance variation.
  2. Please, indicate the number of trainable parameters for the readout layer.
  3. Could the authors speculate on how one can construct a physical reservoir using Drosophila?
  4.  Some other recent physical RCS implementations could be discussed in the introduction part (see e.g. 10.3390/biomimetics8020189, 10.3390/nano13182603, 10.3390/polym17091178)

Author Response

(The authors gave the same response as above.)

Round 2

Reviewer 1 Report

Comments and Suggestions for Authors

The authors have responded appropriately to the reviewer’s comments and have provided an excellent response letter. However, the optimality of the spectral radius used for both the connectome-based and classical reservoirs has not been verified. Figure 13 illustrates performance differences, but does not confirm optimality. A spectral radius of 0.99 may not be optimal, particularly for non-random reservoir computing, and should be adjusted depending on the task. The authors should clearly state that a spectral radius of 0.99 yielded better performance than at least 0.25, 0.5, and 0.75 in both models.
